# HIGH FIDELITY SPEECH SYNTHESIS WITH ADVERSARIAL NETWORKS

**Mikołaj Bińkowski**[*]
Department of Mathematics
Imperial College London
mikbinkowski@gmail.com

**Jeff Donahue, Sander Dieleman, Aidan Clark, Erich Elsen, Norman Casagrande,
Luis C. Cobo, Karen Simonyan**
DeepMind
{jeffdonahue,sedielem,aidanclark,eriche,ncasagrande,
luisca,simonyan}@google.com

## ABSTRACT

Generative adversarial networks have seen rapid development in recent years and have led to remarkable improvements in generative modelling of images. However, their application in the audio domain has received limited attention, and autoregressive models, such as WaveNet, remain the state of the art in generative modelling of audio signals such as human speech. To address this paucity, we introduce GAN-TTS, a Generative Adversarial Network for Text-to-Speech. Our architecture is composed of a conditional feed-forward generator producing raw speech audio, and an ensemble of discriminators which operate on random windows of different sizes. The discriminators analyse the audio both in terms of general realism, as well as how well the audio corresponds to the utterance that should be pronounced. To measure the performance of GAN-TTS, we employ both subjective human evaluation (MOS – Mean Opinion Score), as well as novel quantitative metrics (Fréchet DeepSpeech Distance and Kernel DeepSpeech Distance), which we find to be well correlated with MOS. We show that GAN-TTS is capable of generating high-fidelity speech with naturalness comparable to the state-of-the-art models, and unlike autoregressive models, it is highly parallelisable thanks to an efficient feed-forward generator. Listen to GAN-TTS reading this abstract at https://storage.googleapis.com/deepmind-media/research/abstract.wav.

## 1 INTRODUCTION

The Text-to-Speech (TTS) task consists in the conversion of text into speech audio. In recent years, the TTS field has seen remarkable progress, sparked by the development of neural autoregressive models for raw audio waveforms such as WaveNet (van den Oord et al., 2016), SampleRNN (Mehri et al., 2017) and WaveRNN (Kalchbrenner et al., 2018). A notable limitation of these models is that they are difficult to parallelise over time: they predict each time step of an audio signal in sequence, which is computationally expensive and often impractical. A lot of recent research on neural models for TTS has focused on improving parallelism by predicting multiple time steps in parallel, e.g. using flow-based models (van den Oord et al., 2018; Ping et al., 2019; Prenger et al., 2019; Kim et al., 2019). Such highly parallelisable models are more suitable to run efficiently on modern hardware.

An alternative approach for parallel waveform generation would be to use Generative Adversarial Networks (GANs, Goodfellow et al., 2014). GANs currently constitute one of the dominant paradigms for generative modelling of images, and they are able to produce high-fidelity samples

---

[*]Work done at DeepMind.

that are almost indistinguishable from real data. However, their application to audio generation tasks has seen relatively limited success so far. In this paper, we explore raw waveform generation with GANs, and demonstrate that adversarially trained feed-forward generators are indeed able to synthesise high-fidelity speech audio. Our contributions are as follows:

- We introduce GAN-TTS, a Generative Adversarial Network for text-conditional high-fidelity speech synthesis. Its feed-forward generator is a convolutional neural network, coupled with an ensemble of multiple discriminators which evaluate the generated (and real) audio based on multi-frequency random windows. Notably, some discriminators take the linguistic conditioning into account (so they can measure how well the generated audio corresponds to the input utterance), while others ignore the conditioning, and can only assess the general realism of the audio.

- We propose a family of quantitative metrics for speech generation based on *Fréchet Inception Distance* (FID, Heusel et al., 2017) and *Kernel Inception Distance* (KID, Bińkowski et al., 2018), where we replace the Inception image recognition network with the Deep-Speech audio recognition network. The code for our metrics is publicly available online[1].

- We present quantitative and subjective evaluation of TTS-GAN and its ablations, demonstrating the importance of our architectural choices. Our best-performing model achieves a MOS of $4.2$, which is comparable to the state-of-the-art WaveNet MOS of $4.4$, and establishes GANs as a viable option for efficient TTS.

## 2 RELATED WORK

### 2.1 AUDIO GENERATION

Most neural models for audio generation are likelihood-based: they represent an explicit probability distribution and the likelihood of the observed data is maximised under this distribution. Autoregressive models achieve this by factorising the joint distribution into a product of conditional distributions (van den Oord et al., 2016; Mehri et al., 2017; Kalchbrenner et al., 2018; Arik et al., 2017). Another strategy is to use an invertible feed-forward neural network to model the joint density directly (Prenger et al., 2019; Kim et al., 2019). Alternatively, an invertible feed-forward model can be trained by distilling an autoregressive model using probability density distillation (van den Oord et al., 2018; Ping et al., 2019), which enables it to focus on particular modes. Such mode-seeking behaviour is often desirable in conditional generation settings: we want the generated speech signals to sound realistic and correspond to the given text, but we are not interested in modelling every possible variation that occurs in the data. This reduces model capacity requirements, because parts of the data distribution may be ignored. Note that adversarial models exhibit similar behaviour, but without the distillation and invertibility requirements.

Many audio generation models, including all of those discussed so far, operate in the waveform domain: they directly model the amplitude of the waveform as it evolves over time. This is in stark contrast to most audio models designed for discriminative tasks (e.g. audio classification): such models tend to operate on time-frequency representations of audio (*spectrograms*), which encode certain inductive biases with respect to the human perception of sound, and usually discard all phase information in the signal. While phase information is often inconsequential for discriminative tasks, generated audio signals must have a realistic phase component, because fidelity as judged by humans is severely affected otherwise. Because no special treatment for the phase component of the signal is required when generating directly in the waveform domain, this is usually more practical.

Tacotron (Wang et al., 2017) and MelNet (Vasquez & Lewis, 2019) constitute notable exceptions, and they use the Griffin-Lim algorithm (Griffin & Lim, 1984) to reconstruct missing phase information, which the models themselves do not generate. Models like Deep Voice 2 & 3 (Gibiansky et al., 2017; Ping et al., 2018) and Tacotron 2 (Shen et al., 2018) achieve a compromise by first generating a spectral representation, and then using a separate autoregressive model to turn it into a waveform and fill in any missing spectral information. Because the generated spectrograms are imperfect, the waveform model has the additional task of correcting any mistakes. Char2wav (Sotelo et al., 2017) uses intermediate vocoder features in a similar fashion.

---

[1] `https://github.com/mbinkowski/DeepSpeechDistances`

## 2.2 GENERATIVE ADVERSARIAL NETWORKS

Generative Adversarial Networks (GANs, Goodfellow et al., 2014) form a subclass of implicit generative models that relies on adversarial training of two networks: the *generator*, which attempts to produce samples that mimic the reference distribution, and the *discriminator*, which tries to differentiate between real and generated samples and, in doing so, provides a useful gradient signal to the generator. Following rapid development, GANs have achieved state-of-the-art results in image (Zhang et al., 2019; Brock et al., 2019; Karras et al., 2019) and video generation (Clark et al., 2019), and have been successfully applied for unsupervised feature learning (Donahue et al., 2017; Dumoulin et al., 2017a; Donahue & Simonyan, 2019), among many other applications.

Despite achieving impressive results in these domains, limited work has so far shown good performance of GANs in audio generation. Two notable exceptions include WaveGAN (Donahue et al., 2019) and GANSynth (Engel et al., 2019), which both successfully applied GANs to simple datasets of audio data. The former is the most similar to this work in the sense that it uses GANs to generate raw audio; results were obtained for a dataset of spoken commands of digits from zero to nine. The latter provides state-of-the-art results for a dataset of single note recordings from various musical instruments (NSynth, Engel et al., 2017) by training GANs to generate magnitude-phase spectrograms of the notes (which can easily be converted to waveforms, up to 64000 samples in length). Neekhara et al. (2019) propose an adversarial vocoder model that is able to synthesise magnitude spectrograms from mel-spectrograms generated by Tacotron 2, and combine this with phase estimation using the Local Weighted Sums technique (Le Roux et al., 2010). Saito et al. (2018) propose to enhance a parametric speech synthesiser with a GAN to avoid oversmoothing of the generated speech parameters. Yamamoto et al. (2019) replace the probabilistic distillation loss in Parallel WaveNet with an adversarial version. Tanaka et al. (2018) use a variant of CycleGAN directly on raw waveforms for voice conversion.

GANs have seen limited application at large scale in non-visual domains so far. Two seconds of audio at 24kHz[2] has a dimensionality of $48000$, which is comparable to RGB images at $128 \times 128$ resolution. Until recently, high-quality GAN-generated images at such or higher resolution were uncommon (Zhang et al., 2019; Karras et al., 2019), and it was not clear that training GANs at scale would lead to extensive improvements (Brock et al., 2019).

Multiple discriminators have been used in GANs for different purposes. For images, Denton et al. (2015); Zhang et al. (2017); Karras et al. (2018) proposed to use separate discriminators for different resolutions. Similar approaches have also been used in image-to-image transfer (Huang et al., 2018) and video synthesis (Saito & Saito, 2018). Clark et al. (2019), on the other hand, combine a 3D-discriminator that scores the video at lower resolution and a 2D-frame discriminator which looks at individual frames. In adversarial feature learning, Donahue & Simonyan (2019) combine outputs from three discriminators to differentiate between joint distributions of images and latents. Discriminators operating on windows of the input have been used in adversarial texture synthesis (Li & Wand, 2016) and image translation (Isola et al., 2017; Zhu et al., 2017).

In parallel with our work, MelGAN (Kumar et al., 2019) used GANs for mel-spectrogram inversion, with multiple discriminators operating at different frequencies. MelGAN enforces the mapping between input spectrogram conditioning and generated audio via an $L_1$ reconstruction loss in discriminator feature space, in contrast with our work, which enforces the mapping adversarially using conditional discriminators. When combined with a Text2mel model, their generator achieves a MOS of 3.7.

## 3 GAN-TTS

### 3.1 DATASET

Our text-to-speech models are trained on a dataset which contains high-fidelity audio of human speech with the corresponding linguistic features and pitch information. The linguistic features encode phonetic and duration information, while the pitch is represented by the logarithmic fundamental frequency $\log F_0$. In total, there are $567$ features. We do not use ground-truth duration and

---

[2] 24kHz is a commonly used frequency for speech, because the absence of frequencies above 12kHz does not meaningfully affect fidelity.

pitch for subjective evaluation; we instead use duration and pitch predicted by separate models. The dataset is formed of variable-length audio clips containing single sequences, spoken by a professional voice actor in North American English. For training, we sample 2 second windows (filtering out shorter examples) together with corresponding linguistic features. The total length of the filtered dataset is 44 hours. The sampling frequency of the audio is 24kHz, while the linguistic features and pitch are computed for 5ms windows (at 200Hz). This means that the generator network needs to learn how to convert the linguistic features and pitch into raw audio, while upsampling the signal by a factor of 120. We apply a $\mu$-law transform to account for the logarithmic perception of volume (see Appendix C).

## 3.2 GENERATOR

A summary of generator G's architecture is presented in Table 2 in Appendix A.2. The input to G is a sequence of linguistic and pitch features at 200Hz, and its output is the raw waveform at 24kHz. The generator is composed of seven blocks (GBlocks, Figure 1a), each of which is a stack of two residual blocks (He et al., 2016). As the generator is producing raw audio (e.g. a 2s training clip corresponds to a sequence of 48000 samples), we use dilated convolutions (Yu & Koltun, 2016) to ensure that the receptive field of G is large enough to capture long-term dependencies. The four kernel size-3 convolutions in each GBlock have increasing dilation factors: $1, 2, 4, 8$. Convolutions are preceded by Conditional Batch Normalisation (Dumoulin et al., 2017b), conditioned on the linear embeddings of the noise term $z \sim \mathcal{N}(0, \boldsymbol{I}_{128})$ in the single-speaker case, or the concatenation of $z$ and a one-hot representation of the speaker ID in the multi-speaker case. The embeddings are different for each BatchNorm instance. A GBlock contains two skip connections, the first of which performs upsampling if the output frequency is higher than the input, and it also contains a size-1 convolution if the number of output channels is different from the input. GBlocks 3–7 gradually upsample the temporal dimension of hidden representations by factors of $2, 2, 2, 3, 5$, while the number of channels is reduced by GBlocks 3, 6 and 7 (by a factor of 2 each). The final convolutional layer with *Tanh* activation produces a single-channel audio waveform.

## 3.3 ENSEMBLE OF RANDOM WINDOW DISCRIMINATORS

Instead of a single discriminator, we use an ensemble of Random Window Discriminators (RWDs) which operate on randomly sub-sampled fragments of the real or generated samples. The ensemble allows for the evaluation of audio in different complementary ways, and is obtained by taking a Cartesian product of two parameter spaces: (i) the size of the random windows fed into the discriminator; (ii) whether a discriminator is conditioned on linguistic and pitch features. For example, in our best-performing model, we consider five window sizes $(240, 480, 960, 1920, 3600 \text{ samples})$, which yields 10 discriminators in total. Notably, the number of discriminators only affects the training computation requirements, as at inference time only the generator network is used, while the discriminators are discarded. However, thanks to the use of relatively short random windows, the proposed ensemble leads to faster training than conventional discriminators.

Using random windows of different size, rather than the full generated sample, has a data augmentation effect and also reduces the computational complexity of RWDs, as explained next. In the first layer of each discriminator, we reshape (downsample) the input raw waveform to a constant temporal dimension $\omega = 240$ by moving consecutive blocks of samples into the channel dimension, i.e. from $[\omega k, 1]$ to $[\omega, k]$, where $k$ is the downsampling factor (e.g. $k = 8$ for input window size 1920). This way, all the RWDs have the same architecture and similar computational complexity despite different window sizes. We confirm these design choices experimentally in Section 5.

The conditional discriminators have access to linguistic and pitch features, and can measure whether the generated audio matches the input conditioning. This means that random windows in conditional discriminators need to be aligned with the conditioning frequency to preserve the correspondence between the waveform and linguistic features within the sampled window. This limits the valid sampling to that of the frequency of the conditioning signal (200Hz, or every 5ms). The unconditional discriminators, on the contrary, only evaluate whether the generated audio sounds realistic regardless of the conditioning. The random windows for these discriminators are sampled without constraints at full 24kHz frequency, which further increases the amount of training data.

More formally, let $\lambda = 120$ denote the frequency ratio between waveform and linguistic features and let $\mathsf{D}_k^\mathsf{C}(\star, *; \theta)$ and $\mathsf{D}_k^\mathsf{U}(\star; \theta')$ be conditional and unconditional discriminator networks parametrized by $\theta$ and $\theta'$, respectively, which downsample the waveform input $\star$ by a factor of $k$.[3] We define conditional and unconditional RWDs as stochastic functions:

$$\mathrm{cRWD}_{k,\omega}(x, c; \theta) = \mathsf{D}_k^\mathsf{C}(x_{\mathsf{j:j}+\omega k}, c_{\mathsf{j}/\lambda:(\mathsf{j}+\omega k)/\lambda}; \theta), \qquad \mathsf{j} \sim \mathcal{U}(\{0, \lambda, 2\lambda, \ldots, N - \omega k\}) \quad (1)$$

$$\mathrm{uRWD}_{k,\omega}(x; \theta') = \mathsf{D}_k^\mathsf{U}(x_{\mathsf{j:j}+\omega k}; \theta'), \qquad \mathsf{j} \sim \mathcal{U}(\{0, 1, \ldots, N - \omega k\}), \quad (2)$$

where $x$ and $c$ are respectively the waveform and linguistic features and $a_{l:r} = (a_l, a_{l+1}, \ldots, a_{r-1})^T$ denotes a vector slice.

The final ensemble discriminator combines 10 different RWD's:

$$\mathrm{RWD}_\omega^*(x, c; \theta^*) = \sum_{k \in \{1,2,4,8,15\}} \mathrm{cRWD}_{k,\omega}(x, c; \theta_k) + \mathrm{uRWD}_{k,\omega}(x; \theta_k'), \qquad \theta^* = \bigcup_k (\theta_k \cup \theta_k'). \quad (3)$$

Algorithm 1 in Appendix D shows pseudocode for computation of RWD$^*$. In Section 5 we describe other combinations of RWDs as well as a full, deterministic discriminator which we used in our ablation study.

### 3.4 Discriminator Architecture

The full discriminator architecture is shown in Figure 2. The discriminators consists of blocks (DBlocks) that are similar to the GBlocks used in the generator, but without batch normalisation. The architectures of standard and conditional DBlocks are shown in Figures 1b and 1c respectively. The only difference between the two DBlocks is that in the conditional DBlock, the embedding of the linguistic features is added after the first convolution. The first and the last two DBlocks do not downsample (i.e. keep the temporal dimension fixed). Apart from that, we add at least two downsampling blocks in the middle, with downsample factors depending on $k$, so as to match the frequency of the linguistic features (see Appendix A.2 for details). Unconditional RWDs are composed entirely of DBlocks. In conditional RWDs, the input waveform is gradually downsampled by DBlocks, until the temporal dimension of the activation is equal to that of the conditioning, at which point a conditional DBlock is used. This joint information is then passed to the remaining DBlocks, whose final output is average-pooled to obtain a scalar. The dilation factors in the DBlocks' convolutions follow the pattern $1, 2, 1, 2, \ldots$ – unlike the generator, the discriminator operates on a relatively small window, and we did not observe any benefit from using larger dilation factors.

## 4 Evaluation

We provide subjective human evaluation of our model using Mean Opinion Scores (*MOS*), as well as quantitative metrics.

### 4.1 MOS

We evaluate our model on a set of 1000 sentences, using human evaluators. Each evaluator was asked to mark the subjective *naturalness* of a sentence on a 1-5 Likert scale, comparing to the scores reported by van den Oord et al. (2018) for WaveNet and Parallel WaveNet.

Although our model was trained to generate 2 second audio clips with the starting point not necessarily aligned with the beginning of a sentence, we are able to generate samples of arbitrary length. This is feasible due to the fully convolutional nature of the generator and carried out using a *convolutional masking* trick, detailed in Appendix A.1. Human evaluators scored full sentences with a length of up to 15 seconds.

---

[3] Here we only require that $\star \in \mathbb{R}^{\omega k}$ and $* \in \mathbb{R}^{(\omega k/\lambda) \times 567}$ for any window size $\omega$. Since we consider fully-convolutional architectures with average pooling after the top layer, $\mathsf{D}_k^\mathsf{C}$ and $\mathsf{D}_k^\mathsf{U}$ do not depend on the value of $\omega$. We describe architecture details of $\mathsf{D}_k^\mathsf{C}$'s and $\mathsf{D}_k^\mathsf{U}$'s for $k \in \{1, 2, 4, 8, 15\}$ in Appendix A.2.

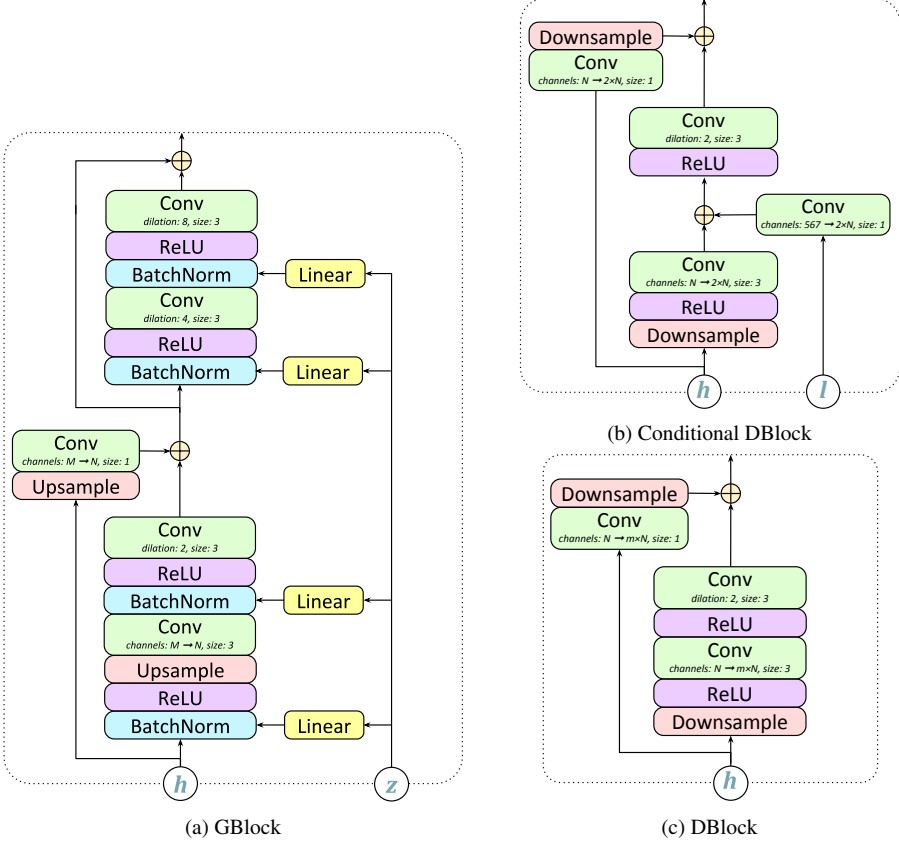

Figure 1: Residual blocks used in the model. Convolutional layers have the same number of input and output channels and no dilation unless stated otherwise. $h$ - hidden layer representation, $l$ - linguistic features, $z$ - noise vector, $m$ - channel multiplier, $m = 2$ for downsampling blocks (i.e. if their downsample factor is greater than 1) and $m = 1$ otherwise, $M$ - G's input channels, $M = 2N$ in blocks 3, 6, 7, and $M = N$ otherwise; *size* refers to kernel size.

## 4.2 SPEECH DISTANCES

We introduce a family of quantitative metrics for generative models of speech, which include the unconditional and conditional *Fréchet DeepSpeech Distance (FDSD, cFDSD)* and *Kernel Deep-Speech Distance (KDSD, cKDSD)*. These metrics follow common metrics used in evaluation of GANs for images, *Fréchet Inception Distance* (FID, Heusel et al., 2017) and *Kernel Inception Distance* (KID, Bińkowski et al., 2018).

FID and KID compute the *Fréchet distance* and the *Maximum Mean Discrepancy* (MMD, Gretton et al., 2012) respectively between representations of reference and generated distributions extracted from a pre-trained *Inception* network (Szegedy et al., 2016). To obtain analogous metrics for speech, we extract the features from an open-source implementation of an accurate speech recognition model, *DeepSpeech2* (Amodei et al., 2016). Specifically, we use the implementation available in the *NVIDIA OpenSeq2Seq* library (Kuchaiev et al., 2018) and extract features from the last layer, whose output is used in the CTC loss during training. We use representations in the resulting feature space to compute the Fréchet distance and MMD (See Appendix B.1 for details).

We note that Kilgour et al. (2019) proposed a similar metric, *Fréchet Audio Distance*. This metric uses a network trained for audio event classification on the AudioSet dataset (Gemmeke et al., 2017) as a feature extractor, whereas we use a network that was trained for speech recognition.

As conditioning plays a crucial role in our task, we compute two variants of these metrics, conditional (cFDSD, cKDSD) and unconditional (FDSD, KDSD). Both Fréchet and Kernel distance provide scores with respect to a *reference real* sample and require both the real sample and the generated one to be independent and identically distributed. Assume that variables $x^{real}$ and $x^{\mathsf{G}}$

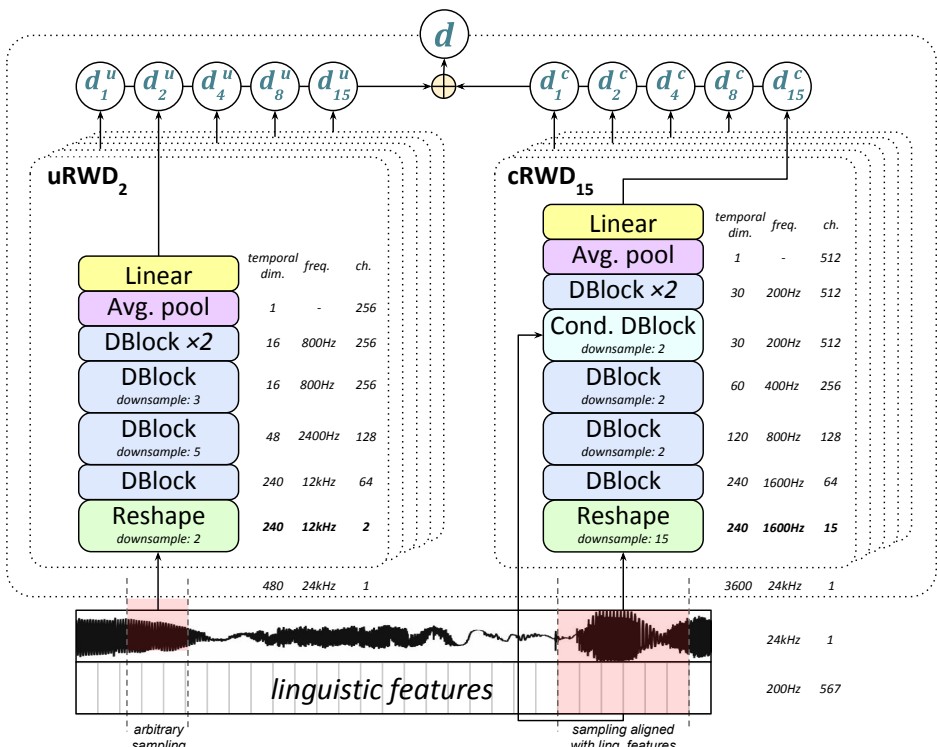

Figure 2: *Multiple Random Window Discriminator* architecture. The discriminator combines outputs from 5 unconditional (uRWDs, left) and 5 conditional (cRWDs, right) discriminators; one of each group is detailed in the diagram. The number of downsampling blocks is fixed for uRWDs and depends on the input window size $\omega k$ for cRWDs, see Table 3. Next to each block we present the dimensionality and frequency of its outputs; *ch.* - number of output channels.

are drawn from the real and and generated distributions, while c is drawn from the distribution of linguistic features. In the conditional case, cFDSD and cKDSD compute distances between conditional distributions $p(\mathrm{x}^G|\mathrm{c})$ and $p(\mathrm{x}^{real}|\mathrm{c})$. In the unconditional case, FDSD and KDSD compare $p(\mathrm{x}^G)$ and $p(\mathrm{x}^{real})$.

Both metrics are estimated using 10,000 generated and reference samples, drawn independently with the same (in the conditional case), or independent (in the unconditional case) linguistic features. This procedure is detailed in Appendix B.3.

The main reason for using both Fréchet and Kernel distances is the popularity of FID in the image domain, despite the issue of its biased estimator, as shown by Bińkowski et al. (2018). Thanks to the availability of an unbiased estimator of MMD, this issue does not apply to kernel-based distances. For instance, they yield zero values for real data, which allows comparison in the conditional case. We give more details on these distances in Appendix B.2.

## 5 EXPERIMENTS

In this section we discuss the experiments, comparing GAN-TTS with WaveNet and carrying out ablations that validate our architectural choices.

As mentioned in Section 3, the main architectural choices made in our model include the use of multiple RWDs, conditional and unconditional, with a number of different downsampling factors. We thus consider the following ablations of our best-performing model:

1. full-input discriminator FullD = $\mathsf{D}_1^\mathsf{C}$,
2. single conditional RWD: cRWD$_1$,

3. multiple conditional RWDs: $\text{cRWD}_{\{1,2,4,8,15\}} = \sum_{k \in \{1,2,4,8,15\}} \text{cRWD}_k$,

4. single conditional and single unconditional RWD: $\text{cRWD}_1 + \text{uRWD}_1$,

5. five independent cRWDs and uRWDs:
   $(\text{cRWD}_1 + \text{uRWD}_1)^{\times 5}(x, c) := \sum_{i=1}^{5} \text{cRWD}_1(x, c; \theta_i) + \text{uRWD}_1(x; \theta_i')$,

6. 10 RWDs without downsampling but with different window sizes:
   $\text{RWD}_{1,240 \times \{1,2,4,8,15\}} = \sum_{k \in \{1,2,4,8,15\}} (\text{cRWD}_{1,240k} + \text{uRWD}_{1,240k})$

7. 10 RWDs with longer window: $\text{RWD}_{480}^*$.

All other parameters of these models were the same as in the proposed one. In Appendix D we present details of the hyperparameters used during training as well as pseudocode for training GAN-TTS.

## 5.1 RESULTS

| model | MOS | FDSD | cFDSD | KDSD $\times 10^5$ | cKDSD $\times 10^5$ |
|---|---|---|---|---|---|
| *natural speech* | $4.55 \pm 0.075$ | 0.161 | N/A | 0 | 0 |
| *WaveNet*, van den Oord et al. (2016) | $4.41 \pm 0.069$ | | | | |
| *Parallel WaveNet*, van den Oord et al. (2018) | $4.41 \pm 0.078$ | | | | |
| FullD | $1.889 \pm 0.057$ | 4.51 | 4.46 | 785 | 782 |
| cRWD$_1$ | $3.394 \pm 0.058$ | 0.362 | 0.247 | 35.2 | 30.9 |
| cRWD$_{\{1,2,4,8,15\}}$ | $3.498 \pm 0.059$ | 0.398 | 0.284 | 42.1 | 37.9 |
| cRWD$_1$ + uRWD$_1$ | $3.502 \pm 0.057$ | 0.259 | 0.144 | 16.6 | 12.3 |
| (cRWD$_1$ + uRWD$_1$)$^{\times 5}$ | $3.526 \pm 0.054$ | 0.194 | 0.073 | 5.59 | 1.34 |
| RWD$_{1,240 \times \{1,2,4,8,15\}}$ | $4.154 \pm 0.050$ | 0.184 | 0.061 | 3.73 | 0.54 |
| RWD$_{480}^*$ | $4.195 \pm 0.045$ | 0.193 | 0.069 | 5.28 | 0.98 |
| GAN-TTS (RWD$^*$) | $4.213 \pm 0.046$ | 0.184 | 0.060 | 3.84 | 0.37 |

Table 1: Results from prior work, the ablation study and the proposed model. Mean opinion scores for natural speech, WaveNet and Parallel WaveNet are taken from van den Oord et al. (2018) and are not directly comparable due to dataset differences. For natural speech we present estimated FDSD – non-zero due to the bias of the estimator – and theoretical values of KDSD and cKDSD. cFDSD is unavailable; see Appendix B.2.

Table 1 presents quantitative evaluations of the proposed model, together with benchmarks and other variants of GAN-TTS that we considered in this work.

Our best model achieves worse yet comparable scores to the strong baselines, WaveNet and Parallel WaveNet. This performance, however, has not yet been achieved using adversarial techniques and is still very good, especially when compared to parametric text-to-speech models. These results are however not comparable due to dataset differences; for instance WaveNet and Parallel WaveNet were trained on 65 hours of data, a bit more than GAN-TTS.

Our ablation study confirms the importance of combining multiple RWDs. The deterministic full discriminator achieved the worst scores. All multiple-RWD models achieved better results than a single cRWD$_1$; all models that used unconditional RWDs were better than those that did not. Comparing 10-discriminator models, it is clear that combinations of different window sizes were beneficial, as a simple ensemble of 10 fixed-size windows was significantly worse. All three 10-RWD models with varying discriminator sizes achieved similar mean opinion scores, with the downsampling model with base window size 240 performing best.

We also observe a noticeable correlation between human evaluation scores (MOS) and the proposed metrics, which demonstrates that these metrics are well-suited for the evaluation of neural audio synthesis models.

## 5.2 DISCUSSION

**Random window discriminators.** Although it is difficult to say why RWDs work much better than the full discriminator, we conjecture that this is because of the relative simplicity of the dis-

tributions that the former must discriminate between, and the number of different samples we can draw from these distributions. For example, the largest window discriminators used in our best model discriminate between distributions supported on $\mathbb{R}^{3600}$, and there are respectively 371 and 44,401 different windows that can be sub-sampled from a 2s clip (real or generated) by conditional and unconditional RWDs of effective window size 3600. The full discriminator, on the other hand, always sees full real or generated examples sampled from a distribution supported on $\mathbb{R}^{48000}$.

**Computational efficiency.** Our Generator has a larger receptive field (590ms, i.e. 118 steps at the frequency of the linguistic features) and three times fewer FLOPs (0.64 MFLOP/sample) than Parallel WaveNet (receptive field size: 320ms, 1.97 MFLOP/sample). However, the discriminators used in our ensemble compare windows of shorter sizes, from 10ms to 150ms. Since these windows are much shorter than the entire generated clips, training with ensembles of such RWDs is faster than with FullD. In terms of depth, our generator has 30 layers, which is a half of Parallel WaveNet's, while the depths of the discriminators vary between 11 and 17 layers, as discussed in Appendix A.2.

**Stability.** The proposed model enjoyed very stable training, with gradual improvement of subjective sample quality and decreasing values of the proposed metrics. Despite training for as many as 1 million steps, we have not experienced model collapses often reported in GAN literature and studied in detail by Brock et al. (2019).

## 6 CONCLUSION

We have introduced GAN-TTS, a GAN for raw audio text-to-speech generation. Unlike state-of-the-art text-to-speech models, GAN-TTS is adversarially trained and the resulting generator is a feed-forward convolutional network. This allows for very efficient audio generation, which is important in practical applications. Our architectural exploration lead to the development of a model with an ensemble of unconditional and conditional *Random Window Discriminators* operating at different window sizes, which respectively assess the realism of the generated speech and its correspondence with the input text. We showed in an ablation study that each of these components is instrumental to achieving good performance. We have also proposed a family of quantitative metrics for generative models of speech: *(conditional) Fréchet DeepSpeech Distance* and *(conditional) Kernel DeepSpeech Distance*, and demonstrated experimentally that these metrics rank models in line with Mean Opinion Scores obtained through human evaluation. The metrics are publicly available for machine learning community, as is the DeepSpeech recognition model they are based on. Our quantitative results as well as subjective evaluation of the generated samples showcase the feasibility of text-to-speech generation with GANs.

ACKNOWLEDGMENTS

We would like to thank Aäron van den Oord, Andrew Brock and the rest of the DeepMind team.

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

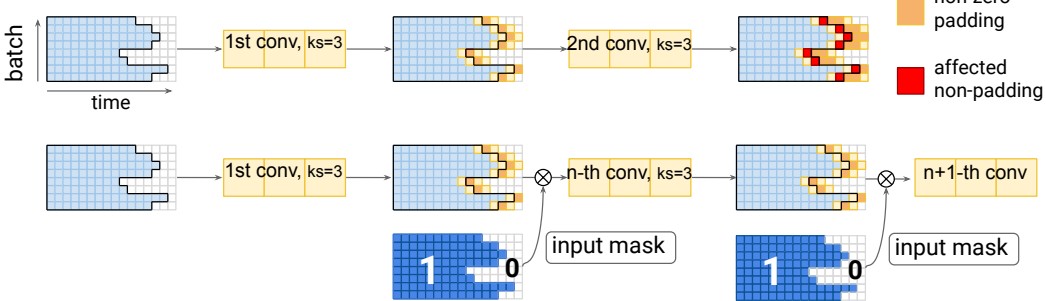

Figure 3: Masking scheme for sampling different-length samples. Top: processing a batch of samples of different lengths padded with zeros leads to interference between padding and non-padding after the second convolution, not seen during training. Bottom: masking after each convolution ensures that the meaningful input seen by each layer is padded with zeros only.

## A    ARCHITECTURE DETAILS

### A.1    MASKING CONVOLUTIONS TO GENERATE LONGER SAMPLES

Since our generator is a fully-convolutional network, in theory it is capable of generating samples of arbitrary length. However, since deep learning frameworks usually require processing fixed-size samples in batches for efficiency reasons, our inputs of different lengths need to be zero-padded to fit in a fixed-size tensor. Convolutional layers, including the ones used in our model, often pad their inputs to create outputs of the desired dimensionality, hence we only need to ensure that the padded part of the input tensors to all layers is always zero. As shown in Figure 3, this would not normally be the case after the second convolutional layer, since convolutions (with kernel sizes greater than one) would propagate non-zero values outside the border between meaningful input and padding. A simple way to address this issue is masking, i.e. multiplying the input by a zero-one mask tensor, directly before each convolutional layer. This enables batched sampling of utterances of different length, which is efficient on many hardware platforms, optimised for batching.

### A.2    GENERATOR AND DISCRIMINATOR DETAILS

| layer/input | $t$ | *freq.* | *ch* |
|---|---|---|---|
| *linguistic features*, $z$ | 400 | 200Hz | 567 |
| conv, *kernel size 3* | 400 | 200Hz | 768 |
| GBlock | 400 | 200Hz | 768 |
| GBlock | 400 | 200Hz | 768 |
| GBlock, *upsample* $\times 2$ | 800 | 400Hz | 384 |
| GBlock, *upsample* $\times 2$ | 1600 | 800Hz | 384 |
| GBlock, *upsample* $\times 2$ | 3200 | 1600Hz | 384 |
| GBlock, *upsample* $\times 3$ | 9600 | 4800Hz | 192 |
| GBlock, *upsample* $\times 5$ | 48000 | 24kHz | 96 |
| conv, *kernel size 3* | 48000 | 24kHz | 1 |
| Tanh | | | |

Table 2: Architecture of GAN-TTS's Generator. $t$ denotes the temporal dimension, while $ch$ denotes the number of channels. The rightmost three columns describe dimensions of the *output* of the corresponding layer.

In Table 2 we present the details of Generator architecture. Overall, the generator has 30 layers, most of which are parts of dilated residual blocks.

Table 3 shows the numbers of residual DBlocks and downsample factors in these blocks for different initial downsample factors of RWDs.

| $k$ | $\mathsf{D}^{\mathsf{C}}_k$ | | | $\mathsf{D}^{\mathsf{U}}_k$ | | |
|---|---|---|---|---|---|---|
| | *factors* | num. blocks | depth | *factors* | num. blocks | depth |
| 1 | $1, 5, 3, 2, 2, 2, 1, 1$ | 8 | 17 | $1, 5, 3, 1, 1$ | 5 | 11 |
| 2 | $1, 5, 3, 2, 2, 1, 1$ | 7 | 15 | $1, 5, 3, 1, 1$ | 5 | 11 |
| 4 | $1, 5, 3, 2, 1, 1$ | 6 | 13 | $1, 5, 3, 1, 1$ | 5 | 11 |
| 8 | $1, 5, 3, 1, 1$ | 5 | 11 | $1, 5, 3, 1, 1$ | 5 | 11 |
| 15 | $1, 2, 2, 2, 1, 1$ | 6 | 13 | $1, 2, 2, 1, 1$ | 5 | 11 |

Table 3: Downsample factors in discriminators for different initial stride values $k$.

All conditional discriminators eventually add the representations of the waveform and the linguistic features. This happens once the temporal dimension of the main residual stack is downsampled to the dimension of the linguistic features, i.e. by a factor of 120. Downsampling is carried out via an initial reshape operation (by a factor $k$ varying per RWD) and then in residual blocks, whose downsample factors are prime divisors of $120/k$, in decreasing order. For unconditional discriminators, we use only the first two largest prime divisors of $120/k$.

Algorithm 1 in Appendix D presents pseudocode for forward pass through RWDensemble.

## B    DEEPSPEECH DISTANCES - DETAILS

### B.1    DEEPSPEECH2

Our evaluation metrics extract high-level features from raw audio using the pre-trained DeepSpeech2 model from the *NVIDIA OpenSeq2Seq* library (Kuchaiev et al., 2018). Let $w = 480$ be a 20ms window of raw audio at 24kHz, and let $f : \mathbb{R}^w \longrightarrow \mathbb{R}^{1600}$ be a function that maps such a window through the DeepSpeech2 network up to the 1600-dimensional output of the layer labeled `ForwardPass/ds2_encoder/Reshape_2:0`. We use default values for all settings of the DeepSpeech2 model; $f$ also includes the model's preprocessing layers.

For a 2s audio clip $\boldsymbol{a} \in \mathbb{R}^{100w}$, we define

$$\mathsf{DS}(\boldsymbol{a}) = \frac{1}{199} \sum_{i=0}^{198} f(\boldsymbol{a}_{iw/2:iw/2+w}) \in \mathbb{R}^{1600},\tag{4}$$

where $\boldsymbol{a}_{i:j} = (\boldsymbol{a}_i, \boldsymbol{a}_{i+1}, \dots, \boldsymbol{a}_{j-1})'$ is a vector slice.

The function DS therefore computes 1600 features for each 20ms window, sampled evenly with 10ms overlap, and then takes the average of the features along the temporal dimension.

### B.2    METRICS IN DISTRIBUTION SPACE

Given samples $\boldsymbol{X} \in \mathbb{R}^{m \times d}$ and $\boldsymbol{Y} \in \mathbb{R}^{n \times d}$, where $d$ is the representation dimension, the Fréchet distance and MMD can be computed using the following estimators:

$$\widehat{\mathrm{Fréchet}^2}(\boldsymbol{X}, \boldsymbol{Y}) = \|\mu_{\boldsymbol{X}} - \mu_{\boldsymbol{Y}}\|_2^2 + \mathrm{Tr}\left(\Sigma_{\boldsymbol{X}} + \Sigma_{\boldsymbol{Y}} - 2(\Sigma_{\boldsymbol{X}}\Sigma_{\boldsymbol{Y}})^{1/2}\right)\tag{5}$$

$$\widehat{\mathrm{MMD}^2}(\boldsymbol{X}, \boldsymbol{Y}) = \frac{1}{m(m-1)} \sum_{\substack{1 \le i,j \le m \\ i \ne j}} k(\boldsymbol{X}_i, \boldsymbol{X}_j) + \frac{1}{n(n-1)} \sum_{\substack{1 \le i,j \le n \\ i \ne j}} k(\boldsymbol{Y}_i, \boldsymbol{Y}_j)$$

$$+ \sum_{i=1}^{m} \sum_{j=1}^{n} k(\boldsymbol{X}_i, \boldsymbol{Y}_j),\tag{6}$$

where $\mu_{\boldsymbol{X}}, \mu_{\boldsymbol{Y}}$ and $\Sigma_{\boldsymbol{X}}, \Sigma_{\boldsymbol{Y}}$ are the means and covariance matrices of $\boldsymbol{X}$ and $\boldsymbol{Y}$ respectively, while $k : \mathbb{R}^d \times \mathbb{R}^d \longrightarrow \mathbb{R}$ is a positive definite kernel function. Following Bińkowski et al. (2018) we use the polynomial kernel

$$k(x, y) = \left(\tfrac{1}{d} x^T y + 1\right)^3.\tag{7}$$

Estimator (5) has been found to be biased (Bińkowski et al., 2018), even for large sample sizes. For this reason, FID estimates for real data (i.e. when $\boldsymbol{X}$ and $\boldsymbol{Y}$ are both drawn independently from the same distribution) are positive, even though the theoretical value of such a metric is zero. KID, however, does not suffer from this issue thanks to the use of the unbiased estimator (6). These properties also apply to the proposed DeepSpeech metrics.

The lack of bias in an estimator is particularly important for establishing scores on real data for conditional distances. In our conditional text-to-speech setting, we cannot sample two independent real samples with the same conditioning, and for this reason we cannot estimate the value of cFDSD for real data, which would be positive due to bias of estimator (5). For cKDSD, however, we know that such an estimator would have given values very close to zero, if we had been able to evaluate it on two real i.i.d. samples with the same conditioning.

### B.3 DISTANCE ESTIMATION

Let $\mathsf{G}$ and $\mathsf{DS}$ represent the generator function and a function that maps audio to DeepSpeech2 features as defined in Eq. 4. Let

$$\boldsymbol{X}^{\mathsf{G}} = \{\mathsf{DS}\left(\mathsf{G}(c_i, z_i)\right)\}_{i=1}^{N}, \qquad \boldsymbol{X}_{:N}^{real} = \{\mathsf{DS}(x_i)\}_{i=1}^{N}, \qquad \boldsymbol{X}_{N:}^{real} = \{\mathsf{DS}(x_i)\}_{i=N+1}^{2N}, \quad (8)$$

where $(x_i, c_i) \overset{iid}{\sim} p(\mathrm{x}^{real}, \mathrm{c}), i = 1, \ldots, 2N$ are jointly sampled real examples and linguistic features, and $z_i \overset{iid}{\sim} \mathcal{N}(0, 1)$. In the conditional case, we use the same conditioning in the reference and generated samples, comparing conditional distributions $p(\mathrm{x}^G|\mathrm{c})$ and $p(\mathrm{x}^{real}|\mathrm{c})$:

$$\widehat{\mathsf{cFDSD}}\left(p(\mathrm{x}^{\mathsf{G}}|\mathrm{c}), p(\mathrm{x}^{real}|\mathrm{c})\right) = \widehat{\mathrm{Fréchet}}\left(\boldsymbol{X}^{\mathsf{G}}, \boldsymbol{X}_{:N}^{real}\right), \quad (9)$$

$$\widehat{\mathsf{cKDSD}}\left(p(\mathrm{x}^{\mathsf{G}}|\mathrm{c}), p(\mathrm{x}^{real}|\mathrm{c})\right) = \widehat{\mathrm{MMD}}\left(\boldsymbol{X}^{\mathsf{G}}, \boldsymbol{X}_{:N}^{real}\right), \quad (10)$$

where $\widehat{\mathrm{Fréchet}}$ and $\widehat{\mathrm{MMD}}$ are estimators of the Fréchet distance and MMD defined in Eq. 5 and 6, respectively.

In the unconditional case, we compare $p(\mathrm{x}^G)$ and $p(\mathrm{x}^{real})$:

$$\widehat{\mathsf{FDSD}}\left(p(\mathrm{x}^{\mathsf{G}}), p(\mathrm{x}^{real})\right) = \widehat{\mathrm{Fréchet}}\left(\boldsymbol{X}^{\mathsf{G}}, \boldsymbol{X}_{N:}^{real}\right), \quad (11)$$

$$\widehat{\mathsf{KDSD}}\left(p(\mathrm{x}^{\mathsf{G}}), p(\mathrm{x}^{real})\right) = \widehat{\mathrm{MMD}}\left(\boldsymbol{X}^{\mathsf{G}}, \boldsymbol{X}_{N:}^{real}\right). \quad (12)$$

## C $\mu$-LAW PREPROCESSING

Many generative models of audio use the $\mu$-law transform to account for the logarithmic perception of volume. Although $\mu$-law is typically used in the context of non-uniform quantisation, we use the transform without the quantisation step as our model operates in the continuous domain:

$$F(x) = \mathrm{sgn}(x)\frac{\ln(1 + \mu|x|)}{\ln(1 + \mu)}, \quad (13)$$

where $x \in [-1, 1]$ and $\mu = 2^8 - 1 = 255$ for 8-bit encoding or $\mu = 2^{16} - 1 = 65,535$ for 16-bit encoding.

Our early experiments showed better performance of models generating $\mu$-law transformed audio than non-transformed waveforms. We used the 16-bit transformation.

## D TRAINING DETAILS

We train all models with a single discriminator step per generator step, but with doubled learning rate: $10^{-4}$ for the former, compared to $5 \times 10^{-5}$ for the latter. We use the *hinge loss* (Lim & Ye, 2017), a batch size of $1024$ and the Adam optimizer (Kingma & Ba, 2015) with hyperparameters $\beta_1 = 0, \beta_2 = 0.999$.

Following Brock et al. (2019), we use spectral normalisation (Miyato et al., 2018) and orthogonal initialisation (Saxe et al., 2014) in both the generator and discriminator(s), and apply off-diagonal

---

**Algorithm 1** TTS-GAN

---

**require:** $N$ - waveform length, $\lambda$ - waveform-conditioning frequency ratio, $\omega$ - base window size,
$\qquad$ $n_{steps}$ - number of training steps, $n_{batch}$ - batch size,
$\qquad$ $\eta_{\mathsf{D}}, \eta_{\mathsf{G}}$ - discriminator and generator learning rates.

1: **procedure** RWD\*$(x, c, \theta^*)$ $\qquad\qquad\qquad\qquad\qquad$ ▷ Compute value of RWDs ensemble
2: $\qquad v \leftarrow 0$
3: $\qquad$ **for** $k \in \{1, 2, 4, 8, 15\}$ **do** $\qquad\qquad\qquad\qquad\qquad\qquad$ ▷ conditional RWDs
4: $\qquad\qquad j \leftarrow \mathcal{U}\left(\{0, \lambda, 2\lambda, \ldots, N - \omega k\}\right)$ $\qquad\qquad\qquad$ ▷ sample random index
5: $\qquad\qquad \bar{x} \leftarrow x_{j:j+\omega k}$ $\qquad\qquad\qquad\qquad\qquad\qquad$ ▷ assign the slice of waveform
6: $\qquad\qquad \bar{c} \leftarrow c_{j/\lambda:(j+\omega k)/\lambda}$ $\qquad\qquad\qquad$ ▷ assign the corresponding slice of conditioning
7: $\qquad\qquad v \leftarrow v + \mathsf{D}_{\mathsf{k}}^{\mathsf{C}}(\bar{x}, \bar{c}; \theta_k)$
8: $\qquad$ **end for**
9: $\qquad$ **for** $k \in \{1, 2, 4, 8, 15\}$ **do** $\qquad\qquad\qquad\qquad\qquad\qquad$ ▷ unconditional RWDs
10: $\qquad\qquad j \leftarrow \mathcal{U}\left(\{0, 1, \ldots, N - \omega k\}\right)$ $\qquad\qquad\qquad$ ▷ sample random index
11: $\qquad\qquad \bar{x} \leftarrow x_{j:j+\omega k}$ $\qquad\qquad\qquad\qquad\qquad\qquad$ ▷ assign the slice of waveform
12: $\qquad\qquad v \leftarrow v + \mathsf{D}_{\mathsf{k}}^{\mathsf{U}}(\bar{x}; \theta_k')$
13: $\qquad$ **end for**
14: $\qquad$ **return** $v$
15: **end procedure**

1: **procedure** TTS-GAN$(\psi, \theta^*)$ $\qquad\qquad\qquad\qquad\qquad\qquad$ ▷ Main training algorithm
2: $\qquad$ **for** $s := 1$ **to** $n_{steps}$ **do**
3: $\qquad\qquad$ #discriminator step
4: $\qquad\qquad ((x_1, c_1), \ldots, (x_{n_{batch}}, c_{n_{batch}})) \overset{iid}{\sim} p(\mathbf{x}, \mathbf{c})$ $\quad$ ▷ sample batch of iid real waveforms and
$\qquad\qquad\qquad\qquad\qquad\qquad\qquad\qquad\qquad\qquad\qquad\qquad\qquad$ corresponding linguistic features
5: $\qquad\qquad c' \leftarrow (c_1', \ldots, c_{n_{batch}}') \overset{iid}{\sim} p(\mathbf{c})$ $\qquad\qquad$ ▷ sample batch of iid linguistic features
6: $\qquad\qquad z \leftarrow (z_1, \ldots, z_{n_{batch}}) \overset{iid}{\sim} \mathcal{N}(0, \boldsymbol{I}_{128})$ $\qquad\qquad\qquad\qquad$ ▷ sample noise
7: $\qquad\qquad x' \leftarrow \mathsf{G}(c', z; \psi)$ $\qquad\qquad\qquad\qquad\qquad$ ▷ generate fake waveforms
8: $\qquad\qquad l_{\mathsf{D}} \leftarrow \frac{1}{n_{batch}} \sum_{i=1}^{n_{batch}} \left((1 - \text{RWD}^*(x_i, c_i, \theta^*))^+ + (1 + \text{RWD}^*(x_i', c_i', \theta^*))^+\right)$
9: $\qquad\qquad\qquad\qquad\qquad\qquad\qquad\qquad\qquad\qquad\qquad\qquad$ ▷ compute Ds' loss
10: $\qquad\qquad \theta^* \leftarrow \text{Adam}(\nabla_{\theta^*} l_{\mathsf{D}}, \eta_{\mathsf{D}})$ $\qquad\qquad\qquad\qquad\qquad\qquad$ ▷ update Ds
11: $\qquad\qquad$ #generator step
12: $\qquad\qquad c' \leftarrow (c_1', \ldots, c_{n_{batch}}') \overset{iid}{\sim} p(\mathbf{c})$ $\qquad\qquad$ ▷ sample batch of iid linguistic features
13: $\qquad\qquad z \leftarrow (z_1, \ldots, z_{n_{batch}}) \overset{iid}{\sim} \mathcal{N}(0, \boldsymbol{I}_{128})$ $\qquad\qquad\qquad\qquad$ ▷ sample noise
14: $\qquad\qquad x' \leftarrow \mathsf{G}(c', z; \psi)$ $\qquad\qquad\qquad\qquad\qquad$ ▷ generate fake waveforms
15: $\qquad\qquad l_{\mathsf{G}} \leftarrow -\frac{1}{n_{batch}} \sum_{i=1}^{n_{batch}} \text{RWD}^*(x_i', c_i', \theta^*)$ $\qquad\qquad$ ▷ compute G's loss
16: $\qquad\qquad \psi \leftarrow \text{Adam}(\nabla_{\psi} l_{\mathsf{G}}, \eta_{\mathsf{G}})$ $\qquad\qquad\qquad\qquad\qquad\qquad$ ▷ update G
17: $\qquad$ **end for**
18: $\qquad$ **return**
19: **end procedure**

---

orthogonal regularisation (Brock et al., 2016; 2019) and exponential moving averaging to the generator weights with a decay rate of 0.9999 for sampling. We also use *cross-replica* BatchNorm (Ioffe & Szegedy, 2015), which aggregates batch statistics from all devices across which the batch is split and *standing statistics* during sampling. The latter means that we accumulate batch statistics from 100 forward passes through the generator before the actual sampling takes place, allowing for inference at arbitrary batch sizes.

In fact, accumulating standing statistics makes the BatchNorm layers in the generator independent of any characteristics of the samples produced during inference. This technique is thus vital for sampling audio of unspecified length: producing samples that are longer than those used during training typically requires using a smaller batch size, with partially padded samples (See Appendix A.1). These smaller batches would naturally have different statistics than the batches used during

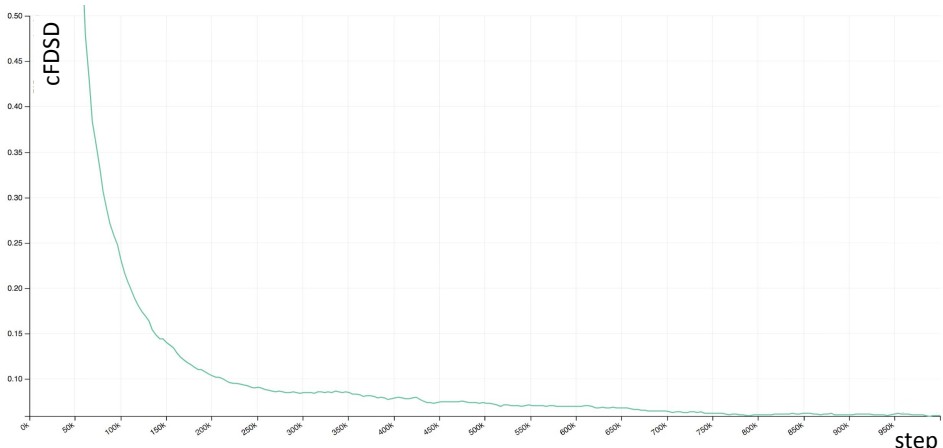

Figure 4: Learning curve for the GAN-TTS model in terms of cFDSD.

training. We trained our models on Cloud TPU v3 Pods with data parallelism over 128 replicas for 1 million generator and discriminator updates, which usually took up to 48 hours.

In Algorithm 1 we present the pseudocode for training GAN-TTS.

Figure 4 presents the stable and gradual decrease of cFDSD during training.

