# OpenReview forum: "High Fidelity Speech Synthesis with Adversarial Networks"
_ICLR.cc/2020/Conference — Accept (Talk)_

### Official Review · AnonReviewer3 · 2019-10-19
**Official Blind Review #3**

**Rating:** 8

**Review:**

I want thank the authors for solving this long-standing GAN challenge in raw waveform synthesis. With all due respect, previous GAN trials for audio synthesis are inspiring, but their audio qualities are far away from the state-of-the-art results. Although the speech fidelity of GAN-TTS is still worse than WaveNet and Parallel WaveNet from the posted sample, it has begun to close the significant performance gap that has existed between autoregressive models and GANs for raw audios. Overall, this is a very good paper with significant contributions to the filed.

Detailed comment:

1, In WaveNet, the conditional features (linguistic / mel-spectrogram) are added as bias terms in the convolutional layers. Did the authors tried this alternative architecture for the generator, which uses the white noisy z as network input (similar as flow-based models, e.g., Parallel WaveNet) and the conditional features as bias term in the convolutional layers?

2, Could the authors comment the importance of serval architecture choices in this work? From Table 1, it seems to me that the ensemble of random window discriminators is the most important (perhaps the only important) contributing factor for the success. For example, the MOS score was boosted from 1.889 to 4.213 by replacing a single full discriminator to the ensemble of RWDs.

3, The notations in Eq. (1) and (2) are messy. Although I can figure their meaning from the context, one may clarify certain notations if they appear at the first time.

4, The stable training (NO model collapses) is pretty impressive. Could the authors shed some light on the potential reason? Does the ensemble of RWD regularizes the training? What's your experience for training FullD (does not have random window ) and cRWD_1 (only has one random window discriminator)? Are they still very stable? Also, could the authors comment on the importance of large batch size -- 1024 for stable training of GAN-TTS?

5, Although there is a notable difference, one may properly mention previous work Yamamoto et al. (2019), which uses GAN as an auxiliary loss within ClariNet and obtains high-fidelity speech ( https://r9y9.github.io/demos/projects/interspeech2019/ ).

Yamamoto et al. Probability Density Distillation with Generative Adversarial Networks for High-Quality Parallel Waveform Generation. 2019.


=== update ===

Thank you for the detailed response.
2,  Thanks for the elaboration.
4,  It would be very interesting to see an analysis of model stability with smaller batch sizes.


**Experience Assessment:**

I have published in this field for several years.

**Review Assessment: Checking Correctness Of Derivations And Theory:**

I carefully checked the derivations and theory.

**Review Assessment: Checking Correctness Of Experiments:**

I carefully checked the experiments.

**Review Assessment: Thoroughness In Paper Reading:**

I read the paper thoroughly.

---

> ### Author Response · Authors · 2019-11-13
> **Response to Official Blind Review #3**
>
> Thank you for the detailed comments.
>
> 1. We did not do experiments with such generator architecture. Although we have considered other architectural choices for generator and ways of conditioning, our early experiments showed that our residual-upsampling scheme is more efficient than parallel wavenet’s full-resolution scheme. The correspondence between temporal dimensions of the conditioning and the waveform also seemed important and hence we decided to keep the proposed generator architecture throughout.
>
> 2. Indeed we believe that the use of the ensemble of random window discriminators was the main factor behind the performance we obtained. This, however, breaks down to three steps:
> (a) switching from full discriminator to random-window discriminator(s),
> (b) including unconditional random window discriminator(s),
> (c) including several different window sizes in the ensemble.
> As can be seen in Table 1., (a) already brings a huge improvement (from ~1.9 to ~3.4 MOS). (b) and (c) also seem to be important; we have considered fixing the window size or using only conditional RWDs, but all of such trials turned out considerably worse. Only models combining all of (a) - (c) made it past MOS of 4.1.
>
> 3. Indeed D^c_k and D^u_k should have been clearly defined there; we clarified this notation in the updated version of the submission.
>
> 4. For the training stability, please see our joint response. As for the role of the batch size, we fixed it throughout all experiments, but we will include analysis of model stability with smaller batch sizes in the final version of the paper.
>
> 5. Thank you for pointing out this related work. We refer to it in the updated version of the submission.

---

### Official Review · AnonReviewer1 · 2019-10-23
**Official Blind Review #1**

**Rating:** 6

**Review:**

This paper proposes to enable GAN based TTS in the time domain with the careful designs of the (non-autoregressive) generator and discriminator. There have been various trials of GAN-TTS but not so many success and I'm glad to hear that the proposed method seems to enable GAN-TTS with fast inference thanks to the non-autoregressive property. The method also proposes new objective measures inspired by the image recognition network based on the high-level features generated by end-to-end ASR, which is also another important contribution of this paper.

My concern for this paper is reproducibility. Although I really appreciate the authors' efforts on providing implementational details in the appendix, the code and data do not seem to be publicly available, and I'm expecting that the implementation of this technique is relatively hard due to their complex designs of the generator and discriminator. Apart from that, the paper is well written overall by well describing the trend of GAN studies in the image processing and the application of such image processing oriented GAN techniques to TTS.

**Experience Assessment:**

I have published one or two papers in this area.

**Review Assessment: Checking Correctness Of Derivations And Theory:**

N/A

**Review Assessment: Checking Correctness Of Experiments:**

I assessed the sensibility of the experiments.

**Review Assessment: Thoroughness In Paper Reading:**

I read the paper at least twice and used my best judgement in assessing the paper.

---

> ### Author Response · Authors · 2019-11-13
> **Response to Official Blind Review #1**
>
> Thank you for your comments. We have added a pseudo-code description of TTS-GAN training algorithm to the updated submission. We believe that, together with other architectural details present in the paper, it makes our work reproducible.

---

### Official Review · AnonReviewer2 · 2019-10-24
**Official Blind Review #2**

**Rating:** 8

**Review:**

This paper puts forth adversarial architectures for TTS. Currently, there aren't many examples (e.g. Donahue et al,  Engel et al. referenced in paper) of GANs being used successfully in TTS, so this papers in this area are significant.

The architectures proposed are convolutional (in the manner of Yu and Koltun), with increasing receptive field sizes taking into account the long term dependency structure inherent in speech signals. The input to the generator are linguistic and pitch signals - extracted externally, and noise. In that sense, we are working with a conditional GAN.

I found the discriminator design very interesting. As the comment below notes, it is a sort of patch GAN discriminator (See pix2pix, and this comment from Philip Isola - https://github.com/junyanz/pytorch-CycleGAN-and-pix2pix/issues/39) and that is could be quite significant in that it classifies at different scales. In the image world, having a single discriminator for the whole model would not take into account local structure of the images. Likewise, perhaps we can imagine something similar in the case of audio at varying scales - in fact, audio dependencies are even more long range. That might be one reason why the variable window sizes work here.

The paper also presents to image analogues for metrics based on FID and the KID, with the features being taken from DeepSpeech2.

I found the speech sample presented very convincing. In general, the architectures are also presented quite clearly, so it seems that we might be able to reproduce these experiments in our own practice. It is also promising that producing good speech could be achieved by a non-autoregressive or attention based architecture.

The authors mention that they hardly encounter any issues with training stability and mode collapse. Is that because of the design of the multiple discriminator architecture?


**Experience Assessment:**

I have read many papers in this area.

**Review Assessment: Checking Correctness Of Derivations And Theory:**

I assessed the sensibility of the derivations and theory.

**Review Assessment: Checking Correctness Of Experiments:**

I assessed the sensibility of the experiments.

**Review Assessment: Thoroughness In Paper Reading:**

I read the paper at least twice and used my best judgement in assessing the paper.

---

> ### Author Response · Authors · 2019-11-13
> **Response to Official Blind Review #2**
>
> Thank you for your comments. Please refer to the joint response in regards to training stability and mode collapse.

---

### Public Comment · ~Rithesh_Kumar1 · 2019-10-11
**Prior work for raw audio generation using Conditional GANs**

We would like to point out that our research paper - MelGAN: Conditional Generative Adversarial Networks for Conditional Waveform Synthesis (accepted as poster presentation at NeurIPS 2019) also performs raw audio generation using generative adversarial networks. Our paper primarily targets the problem of mel spectrogram inversion using Conditional GANs and also show that alternate representations such as VQ-VAE latents, Universal Music Translator encodings can be utilized to generate corresponding raw waveform.

MelGAN and the current paper under review (GAN-TTS) have many similarities in their approach. Specifically, both papers use highly similar Generator architectures (residual blocks, dilated convolutions, pattern of upsampling the conditioning information) and Discriminator architectures (multi-scale discriminator and multiple discriminators, patch-discriminator vs random window samping). The difference occurs in the task, where the GAN-TTS model uses text features instead of mel-spectrograms to perform raw audio generation.

We acknowledge that the authors couldn’t have known this paper since it wasn’t public. It would be nice if the authors could summarize and discuss the additional insights provided by this paper in the light of the existence of this prior work.

We temporarily share the paper using a google drive link, pending arxiv submission. (https://drive.google.com/file/d/1a_CnqAMkFYEC7pfAkBKvjMeaKiREKPkl/view?usp=sharing)

The final camera ready version of the paper will be available later this month (October 30).

---

> ### Author Response · Authors · 2019-10-14
> **Thanks for reference to parallel work**
>
> Thank you for sharing your related work. As it was made public after our submission and will be published only in the near future, it cannot be considered prior work. We are looking forward to reading the camera-ready version of your paper, and will include a discussion of similarities and differences in a future version of our paper.

---

### Author Response · Authors · 2019-11-13
**Joint response to all Reviewers**

We would like to thank all reviewers for their effort and their useful comments.

We have updated our submission, adding several references to related work and pseudocode for training GAN-TTS in Appendix D.

*Stability and Mode Collapse*
There are two phenomena in GAN training: (i) mode collapse and (ii) model collapse. The first manifests itself in the lack of sample diversity, the second is essentially training instability.
In the paper, we didn't claim that our model doesn't have the former (mode collapse). In fact, for conditional generative models like text-to-speech, mode collapse is not necessarily a problem. Having said that, based on our subjective assessment, feeding different noise z samples leads to slightly different speech samples, so the model does capture some sample diversity given the conditioning.
What we did claim in Section 5.2 is that we didn't observe the second phenomenon (model collapse), i.e. training is stable. We attribute this to data augmentation, both explicit - due to training on random crops, and implicit - through discriminating random windows. The only setting in which we observed model collapse was the one with full-window discriminator; settings with even single random window discriminator, on the other hand, led to stable training.

---

### Comment · Area_Chair1 · 2019-11-14
**Reviewers, any comments on the author responses?**

Dear Reviewers, thanks for your thoughtful input on this submission!  The authors have now responded to your comments.  Please be sure to go through their replies and revisions.  If you have additional feedback or questions, it would be great to get them this week while the authors still have the opportunity to respond/revise further.  Thanks!

---

### Decision · Program_Chairs · 2019-12-19

**Decision:**

Accept (Talk)

**Comment:**

The authors design a GAN-based text-to-speech synthesis model that performs competitively with state-of-the-art synthesizers.  The reviewers and I agree that this appears to be the first really successful effort at GAN-based synthesis.  Additional positives are that the model is designed to be highly parallelisable, and that the authors also propose several automatic measures of performance in addition to reporting human mean opinion scores.  The automatic measures correlate well (though far from perfectly) with human judgments, and in any case are a nice contribution to the area of evaluation of generative models.  It would be even more convincing if the authors presented human A/B forced-choice test results (in addition to the mean opinion scores), which are often included in speech synthesis evaluation, but this is a minor quibble.